# Effluent Water Reuse Possibilities in Northern Cyprus

**Gozen Elkiran [1], Fidan Aslanova [2] and Salim Hiziroglu [3,*]**

[1]   Department of Civil Engineering, Near East University, Near East Boulevard, Nicosia 99138,
      Northern Cyprus; gozen.elkiran@neu.edu.tr
[2]   Department of Environmental Engineering, Near East University, Near East Boulevard, Nicosia 99138,
      Northern Cyprus; fidan.aslanova@neu.edu.tr
[3]   Department of Natural Resource Ecology and Management, Oklahoma State University, Stillwater,
      OK 74078, USA
[*]   Correspondence: salim.hiziroglu@okstate.edu; Tel.: +1(405)744-4853

**Abstract:** Northern Cyprus (NC) is suffering from limited water resources and reiterated drought condition experiences due to global warming effects. Previous studies revealed that the water management policy in the country is not sustainable from the perspective of demand and balance. Apparently, the reuse of recycled water will be an alternative resource and can be utilized for some specific purposes to reduce water extraction from the ground. It is expected that treated wastewater will reach 20 million cubic meters (MCM) per year after the completion of the new sewage system for Lefkosa. Today, 20,000 m$^3$ of wastewater is treated at the Lefkosa Central Treatment Plant up to the secondary treatment level, in which the degree of treatment varies from 60% to 95% owing to the weather conditions in the country during the year. Effluent water reuse in NC was not accepted due to cultural belief. However, water scarcity was experienced in the country during the last decade, forcing the farmers to benefit from the recycled water. There is no regulatory framework available in the country for effluent water reuse. However, preparation studies are almost finalized after discussions among government and European Union (EU) agencies. Cyprus, as an EU country, has an obligation to treat the wastewater up to the secondary level before releasing it in an environmentally friendly nature, following the Directive 91/271/EEC. This paper analyzes the effluent water reuse possibilities as a component of integrated water resource management in Northern Cyprus considering laboratory experiment results. It appears that applying tertiary treatment in Northern Cyprus will allow 20 MCM of water contribution to the water budget and it will help protect the vulnerable environment. Also, since the cost of tertiary treatment will be 0.2 United States dollars (USD)/m$^3$, it would be reasonable to prefer this process to the desalination of water, which costs of 1 USD/m$^3$.

**Keywords:** wastewater reuse (WR); Northern Cyprus (NC); water scarcity; alternative water resource (AWR); EU directive (EUD); integrated water resource planning and management (IWRPM)

## 1. Introduction

Northern Cyprus (NC) covers an area of 3355 km$^2$, approximately one-third of Cyprus Island. Nearly half of the coastline of the island is also within the boundaries of NC (Figure 1). NC has a population of approximately 260,000 inhabitants and a population of 300,000 livestock [1]. The water shortfall in almost all countries was aggravated in the last few decades due to population increase, drought condition effects and a decrease in water quality, climate change, and a consequent considerable reduction in rainfall, which makes the purification of water important [2–4]. The over-abstraction of fresh water from groundwater resources caused a high degree of salinization in coastal aquifers and a complete depletion in the interior level [5,6]. To overcome this problem, 41 dams were constructed, of which 18 serve the agricultural sector and the remaining recharge the aquifers in

the regions they are located. The diversion of surplus water from the wet regions to supply recharge water to the aquifers was already applied in the Lefkosa Main Region with a 20-km-long surface flow, despite the high evaporation effects. Small- and moderate-scale desalination projects are applied in some of the regions having shortfall in the country, which are considered to be more reliable resources independent from weather conditions with a cost of about 1 United States dollar (USD)/m$^3$ [6,7]. There are 20,000 m$^3$/day of wastewater flow to the Lefkosa Central Treatment Plant (LCTP), of which 12,000 m$^3$ comes from the southern part of Cyprus, and it is treated up to the secondary level. The level of treatment varies from 60% to 90% depending on the weather conditions [1]. In the summer, due to high evaporation effects (1000 mm/year), the quantity of wastewater is reduced, which makes the treatment process more difficult. In the meantime, in the winter, owing to excessive storm water conditions, the plant cannot treat the wastewater up to the desired level due to limited capacity. Lefkosa has a coverage area of 30 km$^2$. The main wastewater collection system involved cesspools until the last decade. However, after the EU development plan, about 60% of the city was managed by a sewage system. It is obvious that the treatment plant will be insufficient after the completion of the sewage system of the city, which is expected to provide 20 million cubic meters (MCM) per year [8]. Lefkosa as a region has 90 km$^2$ of agricultural land growing vegetables and fruits including artichoke, lettuce, cabbage, leek, apple, apricot, and peach. Currently, the irrigation systems are modified, and sprinkle or drip irrigation systems are used depending on the type of cultivated crop [6,8]. The aquifer located at the area is rather small, with about 1 MCM of storage in wet and 0.5 MCM in drought seasons. Water shortage experienced is around 0.5 MCM in dry periods; however, the water shortage at the nearby region at Guzelyurt is 40 MCM per year [6,8]. Lefkosa as a part of Cyprus, which is an EU member, has an obligation to satisfy the Directive 91/271/EEC, concerning urban wastewater treatment (1991), as amended by Commission Directive 98/15/EC (1998) for environmental protection and upgrade [9]. Accordingly, Member States have to identify the agglomerations with more than 2000 population equivalence (PE). Evidently, the directive will force the members to establish and complete new sewage systems in these cities, which will cause the collection of a substantial amount of wastewater. This concentrated wastewater must somehow be treated up to the secondary level before disposing it to sensible areas, as enforced by the EU Directive [10]. Since the water deficiency shows an increasing trend, effluent water reuse in the agricultural sector is currently focused upon in order to reduce the water stress in the aquifers [11,12]. Furthermore, after the completion of the sewage systems of the cities, wastewater reuse will contribute about 20 MCM to the water supply in Lefkosa in 2025, and reduce the use of fresh water for cultivating the lands [12]. The economic studies on wastewater reuse revealed that tertiary treatment cost is less than the cost of desalinated water, indicating the necessity for the reuse of effluent water as a more reliable water resource [13]. It is estimated that the marginal cost of tertiary treatment effluent is 0.2 USD/m$^3$, while the late desalination cost in NC is about 1 USD/m$^3$. Obviously, this treated effluent water may be used in the agricultural sector for some specific purposes, especially in drought seasons when safe yields of the aquifers are limited due to a remarkable decline in the recharge process [14,15].

The water resources in NC are groundwater resources, semi-perennial low-discharge springs, several surface water reservoirs, and few small-scale desalination plants with a few small-scale sanitary treatment plants (Table 1) [6,16].

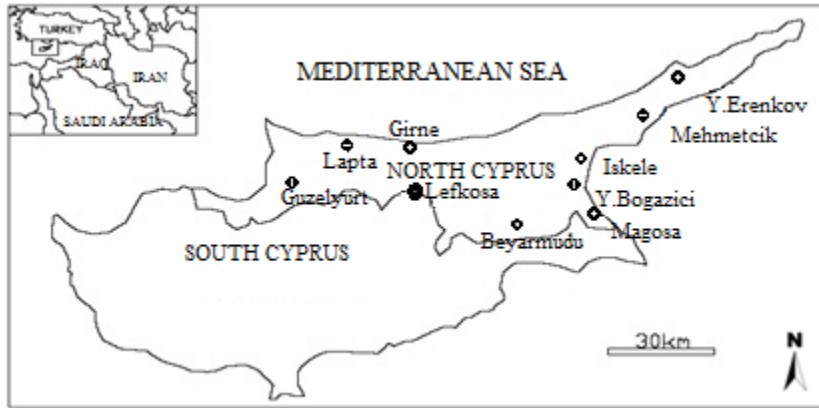

**Figure 1.** Geographical location of Northern Cyprus (NC).

**Table 1.** Total available water at the resources of Northern Cyprus (NC) (Water Works Department, 2016).

| Northern Cyprus | Resource | Volume | Percentage |
|---|---|---|---|
| | Springs | $0.3 \times 10^6$ m$^3$ | 0.3% |
| | Dams | $20.0 \times 10^6$ m$^3$ | 20.4% |
| | Groundwater | $74.1 \times 10^6$ m$^3$ | 75.5% |
| | Desalination | $3.7 \times 10^6$ m$^3$ | 3.8% |
| Total water volume | | $98.1 \times 10^6$ m$^3$ | 100% |

LCTP collects nearly 20,000 m$^3$/day of sewage water from the north and south parts of the city and treats it up to the secondary level. Several local treatment plants are also available as institutional bases. The treated effluent wastewater is used for the irrigation of green fields only, which is about 0.1 MCM [17,18]. To overcome the water shortage, water transportation from Turkey was attempted between 1998 and 2002, with the objective of supplying 7 MCM of water per annum through balloons (so-called Medusa bags). Unfortunately, this system was unsuccessful since only 4.1 MCM of water was transported in five years [8]. Rainfall data obtained from the department of meteorology at the North between 1974 and 2003 was analyzed using a two-year moving average method, whereby a decrease of 2.2 mm/year in rainfall was observed [18]. This catastrophe caused a decrease in the recharging process of the underground resources, resulting in an aggravation of water deficiencies of the aquifers, as experienced in all European countries and, in general, the world [19–21].

## 2. Materials and Methods

The average monthly temperature variations based on districts of NC, for the years from 1975 to 2001, are illustrated in Figure 2. Based on the years between 1975 and 2001, the maximum average temperature was measured in the Guzelyurt region in July as 36.6 °C and its minimum average was 4.6 °C. Similarly, for the abovementioned period, the average temperature distribution in the regions were 18.8 °C in Guzelyurt, 19.3 °C in Lefkosa, 19.9 °C in Girne, 19.8 °C in Magosa, and 19.2 °C in Y. Erenkoy [22].

Monthly rainfall distribution of the country varied considerably among the regions (Figure 3). The minimum annual average value measured in the Lefkosa region was 294.7 mm/year, and the maximum average in the Girne region was 456.6 mm/year. The annual average rainfall for NC was 373.3 mm/year. This figure implies that, for the whole country, only two of the regions observed rainfall more than the average value (Karpaz and Girne), while the others were below the average, indicating drought regions [22].

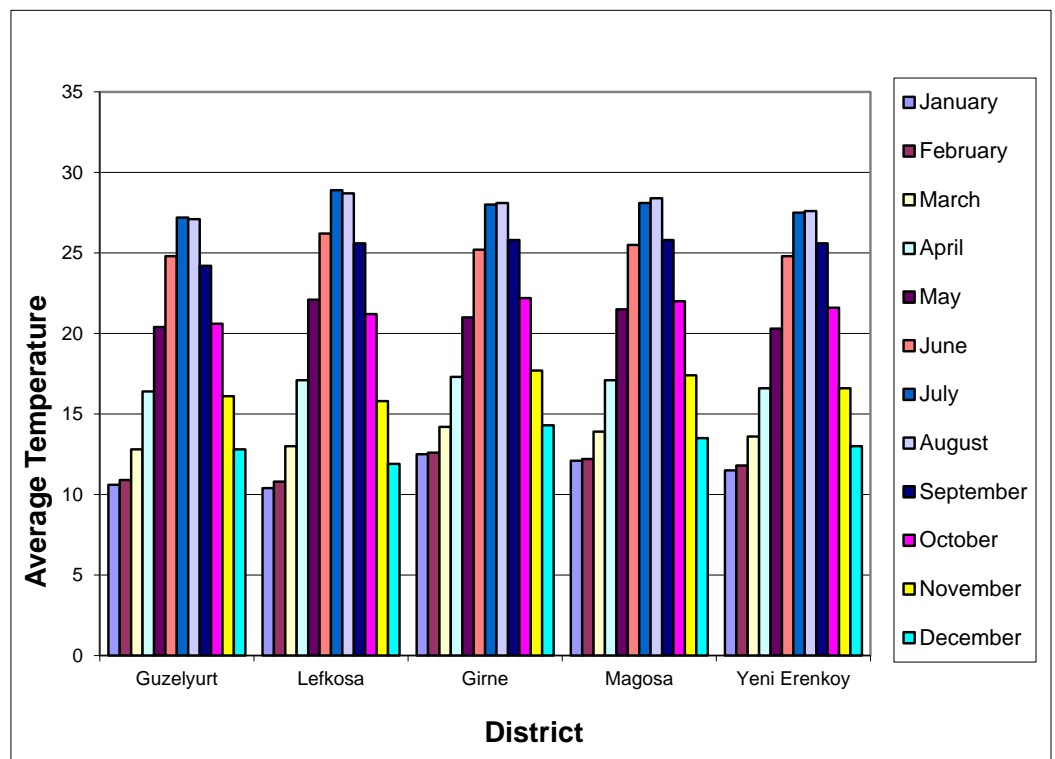

**Figure 2.** Average monthly temperature variations based on districts of NC, in the years from 1975 to 2001 (°C).

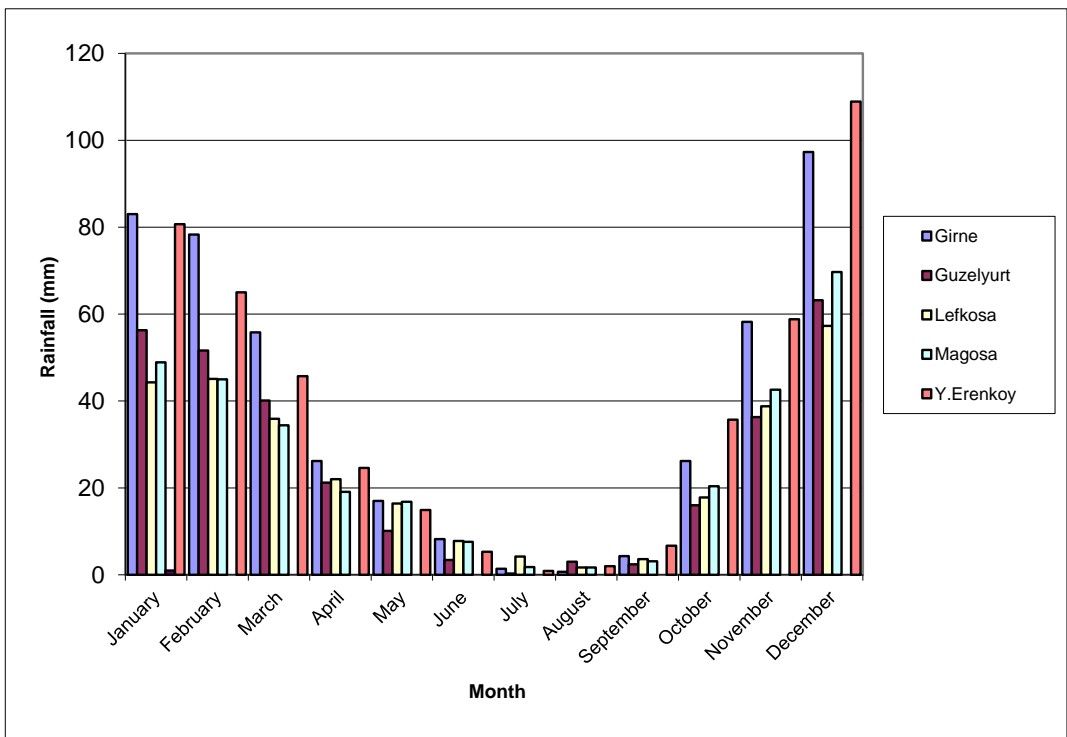

**Figure 3.** Monthly average rainfall values based on different regions of NC between 1975 and 2001.

This study considered the possibility of using recycled water for proper water resource management, to reduce groundwater extraction and for the sustainability of water resources in terms of demand and supply points of view, in the four largest cities of NC including Guzelyurt, Girne, Lefkosa,

and Magusa (Figure 1). This was done by firstly determining the present situation of water resources in NC by analyzing the water balance of the main Magusa coastal, Girne karstic, and Guzelyurt coastal aquifers and the remaining small aquifers in NC (Figure 4). Due to a mismanagement of the Magusa aquifer, it is out of use owing to high salt-water contamination. The Girne karstic aquifer receives the highest precipitation due to its mountainous topography; however, due to steep slopes, most of the water reaches the sea in a short period of time. Roughly, the annual safe yield is 10 MCM and is in balance in terms of demand and supply [16]. The Guzelyurt coastal aquifer is the biggest aquifer with a storage area of 280 km$^2$. Citrus fruit plantation is mainly carried out in this region. The cultivated area dropped from 70 km$^2$ to 55 km$^2$ within the last 25 years, depleting the aquifer 55 m below mean sea level as a result of overharvesting. The NaCl concentration reaches above 5000 ppm along a band of more than 1 km from the coast. The estimated annual safe yield of the aquifer is about 37 MCM [16,17]. Secondly, the concept of integrated water resource planning and management (IWRPM) was applied to estimate the historical water extractions and the consequent deficiencies in the aquifers. Finally, a sophisticated tertiary wastewater treatment process that could efficiently treat the recycled water with less cost is proposed.

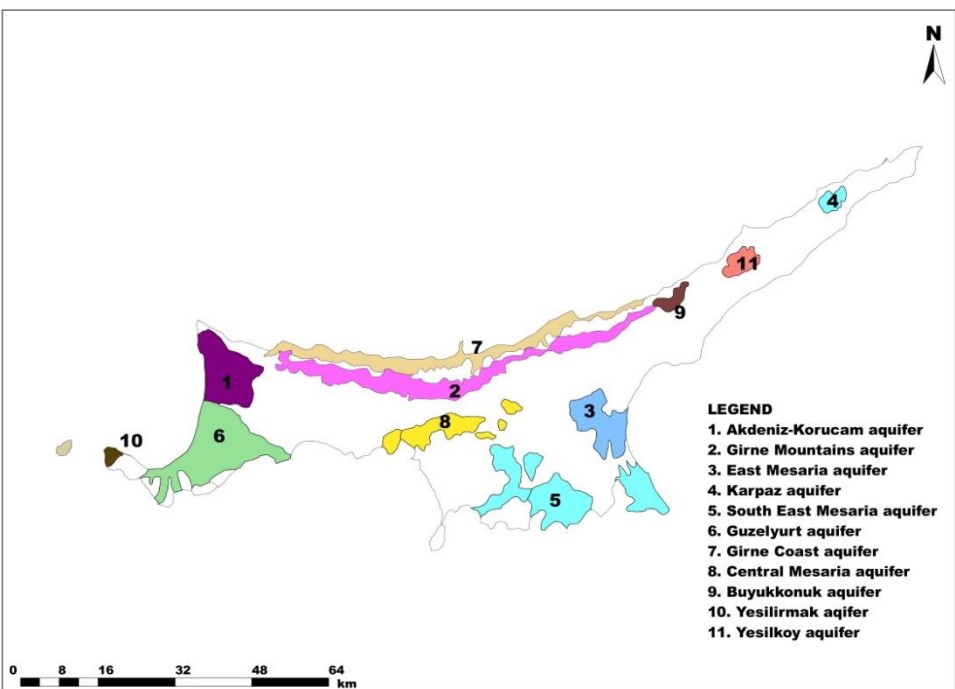

**Figure 4.** Schematic map of aquifers and their locations in NC (Department of Geology and Mines, 2018).

*Governance Systems in Natural Resource Management*

Natural resource management (NRM) is generally defined in this field as the scientific application of ecological knowledge to resource management [23]. Technologies, procedures, and regulations are the main focus areas of this discourse [24]. However, the success of sustainable development is determined by potential relationships among the involved competing actors with natural resources across the entire levels and management structures in different governance scales. To meet competing socioeconomic demands, natural resources are governed, while environmental sustainability is also given priority [25].

## 3. Results and Discussion

Groundwater is the main source of water resources, and provides about 75.5% of the total available water resources in NC, as displayed in Table 1. This reveals that a shortage of water recharge into the aquifers would intensify water scarcity in NC. Therefore, the excessive exploitation of groundwater might lead to a large drawdown of water, which could, in turn, make NC vulnerable to water scarcity for a long period of time. Water recharge into the aquifer takes place slowly and can take several years until it maintains its initial level before drawdown, and a large drawdown would increase the number of years. The agricultural sector consumes the major portion of water resources in NC; by supplying the recycled treated water into agriculture, the water that was initially supplied could be channeled for other domestic use, thereby reducing the groundwater exploitation.

As an EU member, NC has an obligation to fulfill the standards enforced by the EU Directive. Hence, the treatment of wastewater up to the secondary level is compulsory for agglomerations before removal to the environment. This water can be used by the agricultural sector at a minimum cost of tertiary treatment which can be less expensive (0.2 USD/m$^3$) than the desalination of water (1 USD/m$^3$).

### 3.1. The Water Balance Situation in the Country

The aquifer capacities (safe yields) and the water balance in the aquifers foreseen by the Department of Geology and Mines (JMD) of NC are summarized in Table 2, indicating almost 30 MCM of deficit in total. The estimation of historical water extraction and the consequent deficiencies in the aquifer were performed using the integrated water resource planning and management (IWRPM) concept in order to evaluate and compare the results proposed by the JMD (Tables 2–4) [22]. In Tables 3 and 4, the share of the tourism sector is 12% of the domestic use of the country. This research reveals that the deficiencies predicted by the department are highly optimistic, and the real extractions are far beyond those expectations, especially in drought conditions, when deficiencies reach the highest levels due to a reduction in the safe yields of the aquifers, as illustrated in Figure 5 [26–28]. In Figure 5, dry and wet conditions were used in order to distinguish the maximum and minimum average of the extractions and deficiencies in the aquifers. Water tariffs for agricultural use significantly vary from 0.04 to 0.15 USD/m$^3$ of water, depending on the water resource and the region in which it is located. No charge is given to irrigation water in rural areas for poverty alleviation [8,16]. Irrigation water quality also shows great variation in NC. The costal aquifers are affected by seawater intrusion and NaCl concentrations reach up to 5000 ppm in some locations, which makes the water useless. Excessive water withdrawals above the safe yield reduce the quality of water and make it not useful for drinking or as irrigation water [18,27]. Currently, purified water in demijohns (19-liter bottles) supplied by private institutions is used for drinking water in homes, costing about 1.7 USD each. In NC, the sectors benefiting from the water supply are the domestic sector and industry, the agricultural sector, and the tourism sector.

**Table 2.** Aquifer capacities and consequent conditions after annual extractions in NC (Department of Geology and Mines (JMD), 2002) [28].

| Aquifers | Safe Yield (million m$^3$) | Withdrawals (million m$^3$) | Situation (million m$^3$) |
|---|---|---|---|
| Guzelyurt | 37 | 57 | −20 (deficit) |
| Yesilirmak | 7.5 | 7.5 | - |
| Girne Mountains | 11.5 | 11.5 | - |
| East Mesaria | 2 | 8.5 | −6.5 (deficit) |
| South East Mesaria | 2.5 | 2.5 | - |
| Central Mesaria | 0.5 | 0.5 | - |
| Yesilkoy | 1.6 | 3 | −1.4 (deficit) |
| Girne Coast | 5 | 5 | - |
| Karpaz | 1.8 | 1.8 | - |
| Akdeniz-Korucam | 2.7 | 2.7 | - |
| Others (spread) | 2 | 2 | - |
| Total | 74.1 | 103 | −28.9 (deficit) |

**Table 3.** Annual water demands by sector and sources of supply in 2010 [27].

| Sector | Resource | Volume | Percentage | Overall (%) |
|---|---|---|---|---|
| Agriculture | Springs | $0.1 \times 10^6$ m$^3$ | 0% | |
| | Dams | $3.4 \times 10^6$ m$^3$ | 3% | |
| | Groundwater | $96.8 \times 10^6$ m$^3$ | 97% | |
| | Total | $100.3 \times 10^6$ m$^3$ | 100% | 71% |
| Domestic | Springs | $0.2 \times 10^6$ m$^3$ | 1% | |
| | Desalination | $3.7 \times 10^6$ m$^3$ | 9% | |
| | Groundwater | $36.6 \times 10^6$ m$^3$ | 90% | |
| | Total | $40.5 \times 10^6$ m$^3$ | 100% | 29% |
| Total water demand | | $140.8 \times 10^6$ m$^3$ | | 100% |

**Table 4.** Annual water demand by sector and principal use in 2010 [26,27].

| Sector | Unit | Volume | Percentage | Overall (%) |
|---|---|---|---|---|
| Agriculture | Irrigation | $60.8 \times 10^6$ m$^3$ | 61% | |
| | Losses | $39.5 \times 10^6$ m$^3$ | 39% | |
| | Total | $100.3 \times 10^6$ m$^3$ | 100% | 71% |
| Domestic | Household | $29 \times 10^6$ m$^3$ | 72% | |
| | Institutions | $1.7 \times 10^6$ m$^3$ | 4% | |
| | Losses | $9.8 \times 10^6$ m$^3$ | 24% | |
| | Total | $40.5 \times 10^6$ m$^3$ | 100% | 29% |
| Total water demand | | $140.8 \times 10^6$ m$^3$ | | 100% |

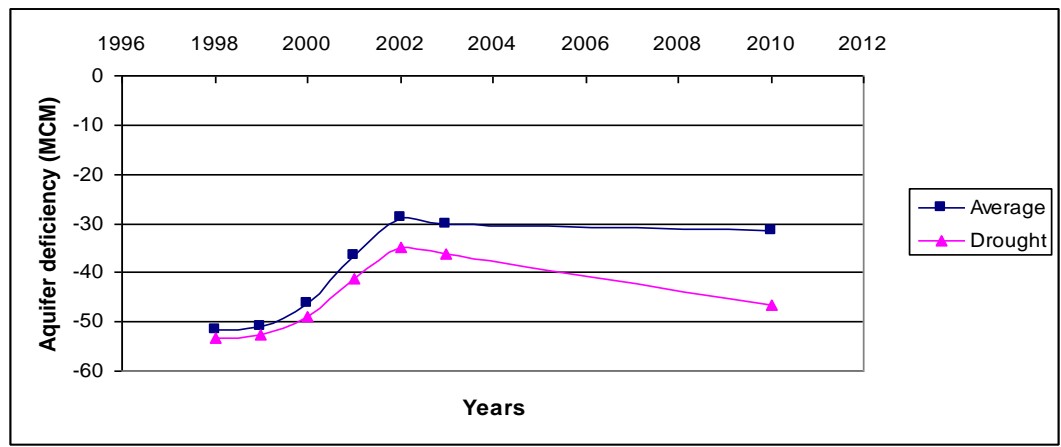

**Figure 5.** Estimated aquifer deficiencies under normal (average) and drought weather conditions owing to excessive water extraction in Northern Cyprus (NC) between 1998 and 2010 [27,28].

*3.2. The Need for Wastewater Reuse in Northern Cyprus*

Previous studies stated that intensive and safe wastewater reuse schemes should be practiced on a large scale in Mediterranean countries due to an increase in the need for water in the irrigation sector [29,30]. Furthermore, the reuse of effluent water to defeat water shortages is one of the components of IWRM, which also contributes to environmental protection and upgrading [31,32]. The tertiary treatment of wastewater offers a cost much lower than the cost of desalinated water, considering that treatment up to the secondary level is unavoidable, according to the Urban Wastewater Treatment Directive (UWTD) 91/271/EEC [9]. UWTD involves the collection, treatment, and disposal of wastewater from agglomerations, and the treatment and discharge of biodegradable wastewater from certain industrial sectors. Since NC is physically part of the island of Cyprus and obtained

EU membership upon reunification of the island, there is an obligation to satisfy the necessities of this directive.

Evidently, the quality of treatment water depends on the intention of use. Domestic use requires water of high quality, whereas agriculture allows lower-quality water. The irrigation of comestible crops necessitates more care because of possible health problems that may be caused by the effluents [14]. Effluent water reuse is a very favorable source in NC and wastewater treatment was initiated in the last two decades; however, its utilization was very limited. Since the treatment degree was not stable and varied from 60% to 90%, the society had negative trust regarding the reuse of effluent water; hence, treated water was diverted into channels upon flowing to the sea. However, recently, farmers living around the plant obligatorily used this effluent water for some specific crop plantations, such as artichoke, eggplant, and maize owing to the water shortage experienced in the last decade [12]. Tertiary treatment was tried in Southern Cyprus, and the cost of production for the tertiary treated effluent was found to be much lower than that for desalinated water. It is estimated that the marginal cost of the tertiary treatment of effluent is currently about 0.2 USD/m$^3$ [14].

### 3.3. Effluent Water Reuse Practices

In NC, four medium-scale central treatment plants were constructed to treat the sewage water of the cities of Girne, Magosa, Guzelyurt, and Lefkosa. The first three of these plants have the capacity to treat up to 600 m$^3$/day. However, the volume treated drops 50% in the winter as the utilization of the freshwater is reduced owing to the colder weather conditions. The Lefkosa Central Treatment Plant (LCTP) treats about 20,000 m$^3$/day of sewage water, of which 60% comes from the northern part and the remainder from the southern part of Lefkosa city [8,28]. There are several small private institutional treatment plants constructed at hotel sites for the irrigation of the lawns, trees, and small gardens in order to reduce the dependence on municipal networks [17]. The decrease in the rainfall intensity in the country resulted in a reduction in the yield capacity of the aquifers due to the inefficient water recharge phenomenon. This caused drops at the water table of the aquifers and started the interaction of wastewater and freshwater. Contamination of the aquifers due to a lack of sewage systems in the cities enforced the government to quicken the installation of new systems for the collection and removal of the wastewater in order to protect the vulnerable groundwater resources. However, the sewage installation process is not yet at the desired level (Table 5) [33]. According to the data obtained by the municipalities in the country, only 15% of sewage systems are completed, which amounts to the 18% of the total population. Currently, the surveying studies for construction are being carried out, allowing the municipalities to arrange new bids for the implementation of the new sewage systems. Guzelyurt city was one of those suffering from the contamination of the freshwater owing to a complete use of the cesspool system for wastewater removal, which was the greatest aquifer supplying water to the whole country [33].

**Table 5.** The completion percentages of the sewage systems in towns in 2008 [22,33].

| City/Town | Completion (%) |
|---|---|
| Girne | 50 |
| Iskele | 40 |
| Mehmetcik | 15 |
| Beyarmudu | 25 |
| Lapta | 100 |
| Y. Bogazici | 25 |
| Lefkosa | 60 |

LCTP is the greatest plant in Northern Cyprus. The first phase of its construction was completed in 1980 over an area of 25 ha. After 20 years, in 2000, the second phase was completed, increasing the capacity up to 20,000 m$^3$/day and the surface coverage area to 75 ha. The plant treats, on average, 4 MCM of wastewater per annum. Treatment phases of the plant are summarized in Figure 6 [26,33]. The

source of the influent of the wastewater is basically domestic. However, some influents from industries are carried by the Lorries and discharged into the system externally. The effluent water treated at the plant is diverted into the Kanlidere Creek to flow out to the sea. Only a limited quantity of this treated water is restricted to use for irrigation purposes (0.1 MCM per year) [8]. The present waste stabilization process requires more area for the ponds in order to be sufficient for future treatment purposes. However, the expansion of the plant is not possible due to the location of the site, which is so close to the residential areas of Lefkosa city. In the meantime, a large pond area also causes very high water loss owing to evaporation effects during summer periods, which can be as much as 10 mm/day. This results in a very limited discharge of recycled water to the creek, and increases the salt concentration of the pond water [34].

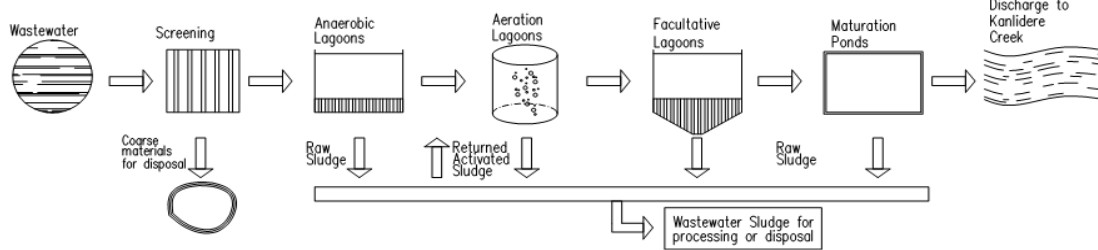

**Figure 6.** Wastewater treatment flowchart of the Lefkosa Central Treatment Plant.

The quality of the treated effluent water of the plant is very poor, since the present stabilization process at the plant is not capable of meeting the limits dictated by UWTD (91/271/EEC) [9] for discharge to sensible areas. The grimy odor affects the whole area around the plant, and spreads to the interior parts of the Lefkosa city. A research concluded by UNOPS revealed that the primary cause of the odors is the biological overloading of anaerobic ponds joined with a high concentration of hydrogen sulfide in the influent [35]. The samples obtained from the LCTP between 2007 and 2008 were analyzed, and the average results obtained from laboratory tests are displayed in Tables 6 and 7 [34].

The electrical conductivity of the treated water measured at the LCTP is considerably high, which makes it difficult to be used for irrigational purposes. This is because of the poor quality of the domestic water supply and very high evaporation effects during the summer, which might reach up to 10 mm/day. According to the tests results, only TSS met the criteria requested by the European Union (EU) directive (Table 8) considering that the population equivalent (PE) is between 2000 and 10,000. This indicates that the treatment degree in the plant is unsatisfactory and further treatment is required. Knowing that, in the near future, the PE will go beyond 10,000, more treatment will be required and the present waste stabilization process will be insufficient; thus, a contemporary treatment system should be introduced. Hence, the government of NC constructed a new plant to serve both the north and south areas with European Union (EU) financial support. The new plant is the largest installation employing membrane bioreactor (MBR) technology, which uses physical, chemical, and organic processes to remove contaminants from the wastewater. The plant has a capacity to produce 10 MCM of water per year, which will be used for irrigation of agricultural lands at the region. The plant produces about three tons of sludge, which can be used as natural fertilizer. Treated water meets the criteria adopted in South Cyprus [36]. This involves the construction of infrastructure for conveyance purposes for the irrigation of cultivated lands, training the beneficiaries using this water so as to not cause any epidemic disease, and acceptance. Great effort was undertaken for the preparation of wastewater reuse standards and criteria with the call of the European Union (EU) and government agencies in last three years. It appears that effluent water will be supplied to the farmers without any charge.

**Table 6.** Experimental results obtained from the Lefkosa Central Treatment Plant between 2007 and 2008.

| Parameters | Inlet Water | Outlet Water | Efficiency (%) |
|---|---|---|---|
| $BOD_5$ (mg/L) | 476 | 30 | 94 |
| COD (mg/L) | 998 | 147 | 85 |
| TSS (mg/L) | 385 | 42 | 90 |
| TP (mg/L) | 18 | 12.8 | 71 |
| TKN (mg/L) | 120 | N/A | N/A |

Note: $BOD_5$: Biological Oxygen Demand; COD: Chemical Oxygen Demand; TSS: Total Suspended Solid; TP: Total Phosphorus; TKN: Total Kjeldahl Nitrogen.

**Table 7.** Additional parameters obtained at the outlet of Lefkosa Central Treatment Plant between 2007 and 2008.

| Parameters | Outlet Water |
|---|---|
| Conductivity micromhos/cm (25 °C) | 3160 |
| F (mg/L) | 0.009 |
| Cl (mg/L) | 540 |
| $NO_2$ (mg/L) | <0.05 |
| Br (mg/L) | 1.40 |
| $NO_3$ (mg/L) | 0.3 |
| $PO_4$ (mg/L) | 38.6 |
| $SO_4$ (mg/L) | 80 |
| $HCO_3$ | 586 |
| Li (mg/L) | <0.001 |
| Na (mg/L) | 371 |
| $NH_4$ (mg/L) | 120 |
| K (mg/L) | 32.2 |
| Fecal coliforms count/100 mL | >2000 |
| Mg (mg/L) | 63.7 |
| Ca (mg/L) | 106.4 |

**Table 8.** Effluent standards required according to the Urban Wastewater Treatment Directive (UWTD) of the European Union (EU).

| Parameters | Maximum Value | Minimum Percentage of Reduction [1] |
|---|---|---|
| | Requirement for normal area | |
| $BOD_5$ | 25 mg/L $O_2$ | 70% to 90% |
| COD | 125 mg/L $O_2$ | 75% |
| TSS | 35 mg/L (WTPs with more than 10,000 PE) 60 mg/L (WTPs with 2000 till 10,000 PE) | 90% (for WTPs with more than 10,000 PE) [2] 70% (for WTPs with 2000 till 10,000 PE) [2] |
| | Additional requirements for sensitive areas | |
| Total P | 2 mg/L P (WTPs with 10,000 till 100,000 PE) 1 mg/L P (WTPs with more than 100,000 PE) | 80% |
| Total N [3] | 15 mg/L N (WTPs with 10,000 till 100,000 PE) [4] 10 mg/L N (WTPs with more than 100,000 PE) [3] | 70% to 80% |

Note: [1] Reduction is in relation to the load of the influent; [2] this requirement is optional; [3] total nitrogen means the sum of Kjeldahl Nitrogen (total organic nitrogen) ($NH_4$-N + organic-N) + $NO_3$-N + $NO_2$-N; [4] these values for concentration are annual means, when the effluent of the biological reactor is superior or equal to 12 °C.

## 4. Conclusions

One of the components of the IWRM is to reuse wastewater in the country as a water resource. Future projections reveal that an enormous quantity of water will be collected from the sewage system of Lefkosa city. Hence, the greatest treatment plant with MBR technology is being constructed in order to treat this quantity of water to a desired level to be used as irrigation water. It is imperative

that the preparation of the standards for the reuse of treated wastewater in the irrigation sector is conducted after the completion of a new plant in order to take effect. Additionally, since no water charge will be given to farmers using effluent water, it will encourage them to use this water with more benefit. Obviously, the quantity of water treated at the LCTP in the near future will be a very reliable alternative water resource with a remarkable quantity (20 MCM). The distribution of this treated water and beneficiaries should be designated and well planned. Desalination of saline water for sensitive areas and for the agricultural sector depending on the type of the crops being grown should be arranged for soil and aquifer protection and efficient crop production, since some of the products are sensitive to the salt content of the irrigation water.

**Author Contributions:** G.E. and F.A. collected the data, conceived the study, and developed the methodology; G.E. and S.H. analyzed and organized the results into figures and tables; G.E., F.A., and S.H. contributed to the manuscript's revision.

**Funding:** This research did not receive any funding.

**Acknowledgments:** The authors acknowledge the support of Northern Cyprus institutions for providing the data used in this study. The authors would like to take the opportunity and thank Taibe Efe Celiker for her support throughout this study.

**Conflicts of Interest:** The authors declare no conflicts of interest.

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
