# Peer review of "Effluent Water Reuse Possibilities in Northern Cyprus"

_water, doi:10.3390/w11020191_

Round 1

Reviewer 1 Report

The article is well given covering in relative detail the subject of the study. Few comments to improve the readability and the quality of the information provided:

L115 In the materials and methods section the first paragraph needs to be more detailed including the methodology and all the data-information used for the purpose of the study. An option is to transfer the basic information provided in L 92-114.Also it might be better to include a table or a figure for precipitation and temperature of the main areas in the materials and methods section.

L125 - L245. This text could be under the Results and Discussion section rather than Materials and Methods. Discussion should be more analytic, It could include some parts that now are presented in the conclusion section.

 L256-L277 Conclusions section should be shortened (see above) including probably only the highlights or bullets. 

Author Response

REVIEWER 1

Title: Effluent Water Reuse Possibilities in Northern Cyprus.

Comments

L115 In the materials   and methods section the first paragraph needs to be more detailed including   the methodology and all the data-information used for the purpose of the   study. An option is to transfer the basic information provided in L   92-114.Also it might be better to include a table or a figure for   precipitation and temperature of the main areas in the materials and methods   section.

We   appreciate respected reviewer’s thoughtful assessment.

Changed   as advised and two figures explaining the weather conditions wereincluded

L125 - L245. This text   could be under the Results and Discussion section rather than Materials and   Methods. Discussion should be more analytic, It could include some parts   that now are presented in the conclusion section.

Changed   as advised

L256-L277 Conclusions   section should be shortened (see above) including probably only the   highlights or bullets. 

The section   was shortened

Reviewer 2 Report

This paper addresses the effluent water reuse possibilities as a component of Integrated Water Resources Management in Northern Cyprus. Although the method and analysis are simple, this work has its practical value. My only concern is that the data presented in this works seem outdated. The authors should provide the most updated information, if there is any. 

Author Response

REVIEWER 2

Title: Effluent Water Reuse   Possibilities in Northern Cyprus.

Comments

This paper addresses the effluent water reuse   possibilities as a component of Integrated Water Resources Management in   Northern Cyprus. Although the method and analysis are simple, this work has   its practical value. My only concern is that the data presented in this works   seem outdated. The authors should provide the most updated information, if   there is any. 

We appreciate respected reviewer’s   thoughtful assessment.

Unfortunately no upto date data is is   available

Reviewer 3 Report

This paper covers an interests thematic area. My main concern is your result and discussion section. This is very limited and not at all enough to elaborate on your research work. You must work again to elaborate the paper widely. Some data in the introduction sections are your materials. They should go Materials and method section. There is no such a section. In the revisions, I highly recommend to read "Developing a Global Compendium on Water Quality Guidelines (2018) very updated. You have to also insert the water quality governance to the paper to see the Cyprus policy changes at a certain level. 

Withanachchi, S.S.; Ghambashidze, G.; Kunchulia, I.; Urushadze, T.; Ploeger, A. A Paradigm Shift in Water Quality Governance in a Transitional Context: A Critical Study about the Empowerment of Local Governance in Georgia. Water 201810, 98.

Author Response

REVIEWER   3

Title: Effluent Water Reuse   Possibilities in Northern Cyprus.

Comments

This paper covers an interests thematic area. My main concern   is your result and discussion section. This is very limited and not at   all enough to elaborate on your research work. You must work again to   elaborate the paper widely. Some data in the introduction sections are your   materials. They should go Materials and method section. There is no such a   section. In the revisions, I highly recommend to read "Developing a   Global Compendium on Water Quality Guidelines (2018) very updated. You have   to also insert the water quality governance to the paper to see the Cyprus   policy changes at a certain level. 

We appreciate respected reviewer’s   thoughtful assessment.

Changed was arranged as advised considering the other   changes advised by the other reviewers.

Revised section    includes  the cited reference.

Reviewer 4 Report

See attached file.

Author Response

REVIEW   4

Title: Effluent Water Reuse   Possibilities in Northern Cyprus.

Author Comments

Line 21‐Northern Cyprus   (NC) repeated: once authors have used NC then go on writing just NC.

Corrected

Line 23‐“Here is no   regulatory framework available in the country for effluent water reuse.”   Maybe authors intend “There is no regulatory framework…”.

Corrected

Line 38‐“Northern Cyprus   (NC) covers had AN area of 3355 km2...”.

Corrected

Line 39‐are authors sure   about the reference 1 related to this sentence?

Ref 1 removed

Line 40-“population of   approximately 260,000 INHABITANTS and having a population of 300,000   livestock”.

Change was done

Lines 40/41‐The estimated   university student population is about 30,000 [1]. I do not see the utility   of this sentence… delete!

Deleted

Line 44‐ over abstraction   should be one word Lines 48/52‐ very confused sentences. Rephrase!

Corrected

Line 54‐“from the   southERN part of Cyprus and IS treated up to secondary level.”

Corrected

Lines 54/55‐ “The level   of treatment varies from 60 to 90 % depending on the weather conditions.”   Explain why!

Explained

Lines 60/63‐ maybe some   commas could make the sentence more readable while there is a “;” too much

Change was arranged and reference   is given

Lines 68/69‐any reference   for those Directives?

Reference was added

Lines 76/78‐confused,   rephrase!

Revised

Line 87‐ “semi perennial   low springs discharge” should be “semi perennial low discharge springs”.

Changed

Figure 1‐ insert all the aquifer   are referred to in text. Insert states names in the geographical small map as   Turkey. Insert all cities are referred to in text or create a new image with   these information.

The cities are inserted into the   map and aquifers are shown in the new figure added

Lines 92/93‐the aquifers   should be shown in the map in figure 1.

Aquifers are shown in the new   figure

Lines 112/114‐“This   catastrophe causes decrease in the recharging process of the underground   resources and as a result, aggravation in the water deficiencies of the   aquifers as it is experienced in the whole European countries [20, 21] AND IN   GENERAL IN THE WORLD [authors should cite some work as Chen, Z.; Grasby, S.E.; Osadetz, K.G.   Relation between climate variability and groundwater levels in theupper   carbonate aquifer, southern Manitoba, Canada. J. Hydrol. 2004, 290, 43–62 AND Water 2017, 9,   788; doi:10.3390/w9100788].

The change was done and the   reference advised was added

Lines 119/124‐ all the   three points mentioned in these lines are very simply resumed in the   following sections of the paper. Keeping reading it is very difficult to   understand what authors’ contribution is because just a water balance is   presented, moreover in a confused way.

Revision was done.

Line 132‐   “Especially, when drought conditions are considered these values reach…” these   values are referred to “deficiencies”? Make it clearer!

Changed

IT IS NECESSARY TO   EXPLAIN HOW THE SAFE YIELD HAS BEEN AVALUATED

Explained. It is given by the   department of Geology and Mines

Table 2‐ the cited   aquifer were just 3… now they are 11! Explain! Aquifers must be shown in map!   Table 3‐ difficult to read! Horizontal lines should help. “Agriculture” has   to stay in caption line?

Explained and horizontal lined   drawn. The three aquifers are the main one and important to the government.   The others are in less importance. Some changes were arranged in the sentence

Table 4‐ same comments as   Table 3. Moreover 4% + 24% for Domestic use do not make 100%!

The mistake was corrected.   Household use was added

Figure 2‐ explain how   those curves have been evaluated. Why to discriminate between average and   drough conditions? Do not authors know if 2002 (for example) was dry or wet?   This part is not clearly explained. Some information more need to be   provided!

Explaination was added.

Line 154‐ “Tertiary   treatment of waste water reuse offers a cost much lower than the   cost…”

corrected

Line 156‐ reference [14]   does not refer to the Directive.

Reference was corrected

Line 158‐ “wastewater   from the certain industrial sectors.” Which sectors?

Explainion was added

Lines 160/163‐ should be   inserted at least before Table 3. Why in Table 3 and 4 the Tourism sector is   not included?

Replaced. Since the country has   very small industry the results are given in the household as a total and it   represents 12 % of the domestic use . This is how the water works department   is evaluating in the country

Lines 167/168‐ Rephrase!

Revised

Line 179/180‐ “... the   volume treated dropS TO 50%...” + “freshwater is reduced” explain why! + LCTP   has not been defined before so insert extended words!

Explained and LCTP was inserted   earlier in the manuscript

Lines 187/193‐ confused!   Rephrase! Table 5‐ cities in the map!

Revised and cities are shown in the   figure

Lines 214/218‐ long and   confused sentence. Rephrase!

Revised

Line 218‐ reference [30]   must stay before reference [31]

Changed

Table 6‐ list acronyms in   extended way in caption

Arranged

Table 7‐horizontal lines   can help the reading

Revised

Table 8‐ horizontal lines   can help the reading.

Revised

Lines 214/218‐ long and   confused sentence. Rephrase!

Rephrased

RESULT AND DISCUSSION   SECTION IS VERY POOR

Improved

CONCLUSIONS ARE VERY   GENERAL

Revised

Round 2

Reviewer 3 Report

The revised version met the all proposed revisions. I would like to accept the paper. 

Author Response

Thanks for your comments.

Reviewer 4 Report

Comments are attached.

Author Response

Responses to the Reviewer #4

Line confusion is because of the track changes applied requested by the editor.

Title: Effluent Water Reuse   Possibilities in Northern Cyprus.

Authors‘    responses

Suggested citation Water 2017, 9, 788; doi:10.3390/w9100788 is missing

References  were added

Line   123, Ref [19] Line 533

Figure 2 modification is needed

Figure   2 was revised to its new format as requested

Figure 4 improve resolution

Resolution   of Fig 4 was improved

Line 357 are authors sure about the reference to  Fig 2 shows average T not safe yield

The   mistake was corrected Line 335

Table 3 and   4: I suggest to insert lines as the green ones indicated below

Lines   inserted as advised

Is still   missing in text an explanation regarding the absence of the tourism sector in   table 3 and 4

Line   about tourism sector was added on  Line   331

Table 8:   horizontal lines still missing. It is very difficult to read the table

Horizontal   lines and additional corrections were added to put the table in a better   format as requested.